# Cre-Recombinase Induces Apoptosis and Cell Death in Enterocyte Organoids

**DOI:** 10.3390/antiox11081452

**Published:** 2022-07-26

**Authors:** Franziska Moll, Manuela Spaeth, Katrin Schröder

**Affiliations:** Institute for Cardiovascular Physiology, Goethe-University, 60590 Frankfurt, Germany; franziska.moll@iba-heiligenstadt.de (F.M.); spaeth@vrc.uni-frankfurt.de (M.S.)

**Keywords:** Cre-recombinase, enterocytes, organoids, apoptosis, cell death, genomic instability

## Abstract

The culture of primary intestinal epithelia cells is not possible in a normal culture system. In 2009 a three-dimensional culture system of intestinal stem cells was established that shows many of the physiological features of the small intestine, such as crypt-villus structure, stem cell niche and all types of differentiated intestinal epithelial cells. These enteroids can be used to analyze biology of intestinal stem cells, gut homeostasis and the development of diseases. They also give the possibility to reduce animal numbers, as enteroids can be cryo-conserved and cultivated for many passages. To investigate the influence of genes such as NADPH oxidases on the gut homeostasis, transgenic approached are the method of choice. The generation of enteroids from knockout mice allows real-time observations of knockout effects. Often conditional knockout or overexpression strategies using inducible Cre recombinase are applied to avoid effects of adaption to the knockout. However, the Cre recombinase has many known caveats from unspecific binding and its endonuclease activity. In this study, we show that although NADPH oxidases are important for *in vivo* differentiation and proliferation of the intestine, their expression is drastically reduced in the organoid system. Activation of Cre recombinase by 4-hydroxy tamoxifen in freshly isolated enteroids, independently of floxed genes, leads to decreased diameter of organoids. This effect is concentration-dependent and is caused by reduced cell proliferation and induction of apoptosis and DNA damage. In contrast, constitutive expression of Cre has no impact on the enteroids. Therefore, reduction of tamoxifen concentration and treatment duration should be carefully titrated, and appropriate controls are necessary.

## 1. Introduction

Animals, such as mice, rats, hamsters and rabbits have been used in research for a long time, with a steady rise in the number. Every year millions of animals are used to understand the effects of medical procedures or to refine them; to test for toxicity of potential new drugs or just to analyze physiological processes [1]. Ethical debates argue that experiments causing pain and distress to the animals are unethical, even up to the demand that animal experiments must be stopped [2]. Accordingly, reduction, refinement and replacement (3R) for experiments using laboratory animals is the goal [3]. On the heels of 3R, tissue engineering, 3D- and organoid-based studies became an attractive alternative. This holds especially true for cells that are hard to culture, such as gut epithelial cells. Generation of intestinal organoids can recapitulate the three dimensional crypt-villus structure of the small intestine [4]. Such a mini-gut culture system as established in 2009, can be performed under serum free and thereby well definable culture conditions [5]. Long term culture and cryo-conservation are possible, which allows the use of enteroids from one mouse for a variety of experiments [6].

In addition to the use of CrispR/Cas9 methods for deletion of specific genes [7], conditional knockouts or transgenic mice allows real-time observations of gene-dependent effects [4]. So far, the most often used system is Cre/loxP, which is based on overexpression of the bacteriophage P1 enzyme Cre-recombinase in all or certain cells of an animal. Both variants exist, those being constitutive and inducible knockouts. For inducible knockouts, activation by an external stimulus enables the enzyme to enter the nucleus. The most prominent way of activation is to feed whole animals with tamoxifen or to supplement a cell culture with 4-hydroxy-tamoxifen (4-OHT). Cre-recombinase dissects parts of a gene or DNA segments flanked by a 34 bp recognition sequence (loxP) consisting of two 13 bp palindromes flanking a 8 bp core sequence [8,9]. Mammalian genomes however, carry endogenous sequences that could be recognized as pseudo loxP sites [10]. About 250 of such pseudo-loxP sites are predicted in the mouse genome [11]. Alteration of the central 8 bp recognition pattern not only impacts the efficiency of Cre-recombination, but importantly truncation of this motif leads to the formation of single stranded nicks in the DNA [12]. In fact, overexpression of Cre-recombinase directly alters genome stability, leading to increased chromatid breaks, dicentric chromosomes, sister chromatid exchange and aberrant gaps or fragments [13]. Physiological consequences of endonuclease activity of Cre-recombinase are decreased cell proliferation, increased apoptosis and cell cycle arrest [14]. Despite the use of self-deleting Cre-recombinases [15] and self-excising retroviral vectors [16] and other systems such as the Vika/vox [11], problems persist.

We hypothesize the following: Livelong exposure to Cre-recombinase will lead to compensatory mechanisms to enable the organisms’ life. In contrast, acute activation of Cre-recombinase induces DNA damage and other harmful effects that may be misinterpreted as effects of the target gene intended to be eliminated.

## 2. Materials and Methods

### 2.1. Mice

All animal procedures were performed in accordance with animal protection guidelines. C57BL/6 mice were purchased from Charles River (Deisenhofen, Germany). Wildtype mice (denoted as Cre0/0) in comparison with differentially floxed CreERT2 mice describe siblings that do not contain the CreERT2 allele. NoxO1_flox_CreERT2 (abbreviated as NoxO1) and NoxO1 constitutive knockout (NoxO1^−^^/^^−^) mice were generated as described elsewhere [17]. Nox4_flox_CreERT2 (abbreviated as Nox4) and constitutive Nox4 knockout (Nox4^−^^/^^−^) mice were characterized before [18]. gtRosa26_CreERT2 and gtRosa26_Cre were kindly provided by Frank Schnütgen and Ingrid Fleming, University of Frankfurt. All animals were bred in the local animal facility.

### 2.2. Small Intestinal Organoids

Crypt isolation and organoid cultures from murine small bowel were performed based on the methods of Mahe and colleagues [6] and using the Intesticult™ system from StemCell Technologies (Vancouver, Canada) according to manufacturer’s instructions. Small intestines were isolated from mice and crypts isolated as described. 300 crypts were seeded in 50 µL mixture of Matrigel and Intesticult™ medium in 24-well plates. For MTT assay 150 crypts were seeded in 25 µL mixture in 48-well plates. Diphenyleneiodonium (DPI) and 4-OHT were added to the medium after seeding in indicated concentrations and removed after 48 h. On day 4 after isolation organoids were analyzed. To isolate organoids from the matrigel and prepare them for further analysis, TrypLE© Express (Life Technologies, Carlsbad, CA, USA) was added to the matrigel domes. Matrigel was digested for 10 min at 37 °C. The reaction was stopped by dilution with Dulbecco’s phosphate-buffered saline (DPBS) (Thermo Fisher Scientific, Waltham, MA, USA) and the organoids were processed for further analysis.

### 2.3. RNAScope^®^ In Situ Hybridization

In situ hybridization by RNAscope^®^ technique was performed according to the manufacturer’s instructions (Advanced Cell Diagnostics (ACD), Newark, NJ, USA) in paraffin embedded sections of duodenum swiss rolls, previously fixed with 4% PFA (paraformaldehyde). Five micrometers thick sections were deparaffinized and treated with H_2_O_2_ followed by antigen retrieval and protease treatment according to kit’s instructions (RNAscope H_2_O_2_ and Protease Plus reagents ACD #322330). Probes to NoxO1 ACD #466541; Nox4 ACD #457261 and positive/negative controls (peptidylprolyl isomerase B ACD #313911/*B. subtilis* dihydrodipicolinate reductase ACD #310043) were hybridized for 2 h followed by 6 amplification steps. The signal was detected with RNAscope 2.5 HD detection kit Brown (ACD #322310) and specimens counterstained with hematoxylin. Slides were analyzed by light microscopy at 10 × magnification.

### 2.4. qRT-PCR

Total mRNA from frozen homogenized tissue was isolated with an RNA-Mini-kit (Bio&Sell, Feucht, Germany) according to the manufacturer’s protocol. Random hexamer primers (Promega, Madison, WI, USA) and Superscript III Reverse Transcriptase (Invitrogen, Darmstadt, Germany) were used for cDNA synthesis. Semi-quantitative real-time PCR was performed with AriaMx qPCR cycler (Agilent Technology, Santa Clara, CA, USA) using iQ™ SYBR^®^ Green Supermix (BioRad, Hercules, CA, USA) with appropriate primers. Relative expressions of target genes were normalized to eukaryotic translation elongation factor 2 (EF2), analyzed by delta-delta-Ct method and represented as percent of control samples. Primer sequences are listed in the following Table 1.

### 2.5. Protein and Western Blot Analysis

For separation of nucleus and cytosol, the cells were lysed in buffer A (10 nM HEPES pH 7.9, 10 nM KCL, 0.1 mM EDTA, 0.1 mM EGTA, 1% Nonidet, 10 mM DTT, protein-inhibitor mix (PIM), 40 µg/mL phenylmethylsulfonylfluorid (PMSF). Cells were centrifuged to gain the cytosol containing supernatant. The pellet was cooked in sample buffer to gain the nuclear fraction. Bradford assay was used to determine the protein amount in the cytosolic fraction [19]. Samples were cooked in sample buffer and were transferred on SDS-PAGE followed by Western Blotting. Identical sample volumes from nuclear and cytosolic fractions were loaded. Analysis was performed with an infrared-based detection system using fluorescent-dye-conjugated secondary antibodies from LI-COR biosciences, Bad Homburg, Germany.

Primary antibodies used are: D7L7L from Cell signaling, Danvers, MA, USA for Cre-recombinase, A1978 from Sigma-Aldrich, St. Louis, MO, USA for β-actin and sc-5342 from Santa Cruz Biotechnology, Dallas, TX, USA for Topoisomerase I.

### 2.6. Cell Counting

After isolating the organoids from the matrigel and generating single cells, the cell number was analyzed using the NucleoCounter^®^ NC-3000™ (Chemometec, Allerod, Denmark) using the cell count and viability assay according to manufactures’ instructions. 

### 2.7. Cell cycle and DNA Fragmentation Assay

DNA fragmentation was analyzed using the NucleoCounter^®^ NC-3000™ (Chemometec, Allerod, Denmark) according to manufacturer’s protocol with minor adjustments. Matrigel was digested using TrypLE™ Express Enzyme (Thermo Fisher Scientific, Waltham, MA, USA) for 15 min at 37 °C. Organoids were manually disrupted and washed using PBS. After at least 12 h of fixation in 70% Ethanol, cells were stained in DAPI (4′,6-Diamidin-2-phenylindol) staining solution (Solution 3, #910-3003, Chemometec, Allerod, Denmark) and finally analyzed using the NC-Slide A8™ (Chemometec, Allerod, Denmark).

### 2.8. Comet Assay

To analyze DNA single and double strand breaks in the organoids, the Comet Assay^®^ Reagent kit from Trevigen (Gaithersburg, MD, USA) was used according to manufacturer’s instructions. The alkaline protocol was used. Electrophoresis was performed at 25 V for 30 min. Pictures were taken with a confocal microscope LSM 800 Meta (Carl Zeiss Microscopy GmbH, Jena, Germany) and quantification was carried out manually determining the ratio of cells with comets/cell number.

### 2.9. Apoptosis Assay-Annexin V Staining

Apoptosis was analyzed using the NucleoCounter^®^ NC-3000™ (Chemometec, Allerod, Denmark) according to manufacturer’s protocol using Annexin V from Biotium (Fremont, CA, USA).

### 2.10. Statistics

Unless otherwise indicated, data are given as means ± standard error of mean (SEM). Calculations were performed with Prism 5.0 (GraphPad Software, San Diego, CA, USA). Individual statistics of unpaired samples was performed by two-tailed *t*-test or one-way Anova with Bonferroni post-test if applicable. Grouped data were analyzed using two-tailed 2-way Anova with Bonferroni post-test. A *p*-value of <0.05 was considered as significant. Unless otherwise indicated, n indicates the number of individual experiments.

## 3. Results

### 3.1. Acute Activation of Cre-Recombinase Hinders Organoid Growth

Here we analyzed two models of transgenic Cre-recombinase in mice: gtRosa26Cre and gtRosa26CreER^T2^. Both strains constitutively overexpress Cre-recombinase in all cells of the body, which is permanently active in gtRosaCre transgenics. In contrast, in gtRosaCreER^T2^ transgenics, activation and nuclear translocation of the overexpressed Cre-recombinase depends on tamoxifen as an external stimulus. Different to whole animals, cells are not able to metabolize tamoxifen. Accordingly, in cell culture 4-OHT is the stimulus of choice, with ethanol serving as solvent and as control. Treatment of freshly isolated gtRosaCreER^T2^-intestine crypts with 1 or even 0.1 µM 4-OHT prompts Cre-recombinase to migrate from the cytosol into the nucleus as verified by Western blot following cell fractionation (**Figure 1A**). This translocation of Cre-recombinase surprisingly impairs proliferation (**Figure 1B**). EtOH treated gtRosaCreER^T2^ crypts grew linearly over a 4 days period, while 4-OHT significantly flattened the curve in a dose dependent manner. If treated with 1 µM 4-OHT the cell number even slightly decreased between day 2 and 4, indicating death of cells. In contrast, crypts isolated from gtRosaCre intestines grew linearly over the observed time. In order to study the effects of 4-OHT and Cre-recombinase in more detail, enterocytes were grown as enteroids (organoids made out of enterocytes). As such, they build 3D structures as depicted in **Figure 1C,D**. Organoid number indicates the ability of the isolated crypts to survive and build organoids, while organoid diameter serves as a proxy for cell proliferation within the organoids. 4-OHT itself had no effect on organoid number or growth, while in Cre-recombinase positive gtRosaCreER^T2^-organoids, treatment with 4-OHT reduced organoid diameter and number in a dose dependent manner (**Figure 1C**). In contrast, number and diameter of gtRosaCre-organoids with constitutive active Cre-recombinase did not differ from the corresponding wildtype (**Figure 1D**).

We conclude that acute 4-OHT induced nuclear translocation of Cre-recombinase prevents enterocyte proliferation in organoids, while constitutive active Cre-recombinase does not.

### 3.2. Mitochondrial Dysfunction and Acute Activation of Cre-Recombinase but Not ROS Inhibit Growth and Survival of Small Intestinal Organoids

Reactive oxygen species (ROS) are thought to be harmful in in many occasions. Accordingly, also for intestine, ROS have been shown to be harmful for proliferation and wound repair [20,21]. Indeed, H_2_O_2_ was able to impair organoid survival and proliferation at concentrations of 500µM (**Figure 2A**). Interestingly, diphenyleneiodonium (DPI), a known inhibitor of ROS formation by flavoproteins, reduced survival and proliferation of organoids as well (**Figure 2A,B**). DPI inhibits all flavoproteins and thereby impairs mitochondrial function and mitochondrial dysfunction has been shown to impair growth and survival of intestine organoids [22]. Reduction of proliferation and survival might be a consequence of inhibition of the pentose phosphate pathway which produces central molecules for the synthesis of desoxyribonucleotides [23]. Other essential proteins inhibited by DPI include cytochrome P450 reductase [24] and mitochondrial complex I [25] and treatment with DPI can induce oxidative stress [23]. Accordingly, the DPI data may be quite unspecific and not related to ROS formation at all.

In addition to mitochondria, NADPH oxidases (Nox) are important flavoproteins and ROS producers. Quantitative PCR revealed that almost all homologues and subunits of NADPH oxidases were expressed in tissue of the small intestine (**Figure 3A**).

The small membrane bound subunit p22phox, which is required for activity of Nox1, Nox2 and Nox4, was most highly expressed. The broad spectrum of p22phox dependencies in NADPH oxidases makes it an unattractive target, if the goal is to study the role of individual ROS producing enzymes. We therefore decided to study the role of the most and least prominent expressed NADPH oxidase subunits, which are NoxO1 and Nox4. NoxO1 was expressed in epithelial cells with stronger expression patterns towards the crypts of the small intestine, whereas Nox4 was not only detected in epithelial cells but also in capillaries and connective tissue (**Figure 3B**). In order to analyze the effect of the individual NADPH oxidase subunits we utilized intestinal organoids from tamoxifen inducible flox_CreER^T2^- and global NoxO1 and Nox4 knockout mice. Treatment with either 0.1 or 1 µM 4-OHT for 48 hours impaired growth of organoids in both floxed CreER^T2^-lines when compared to the ethanol-treated controls (**Figure 3C**). Those data fit very well to the finding that DPI and thereby reduction of ROS prevents survival and growth of organoids. A conclusion at this point could be that a reduction of NADPH oxidases and subsequent ROS formation will prevent organoid growth. Importantly, number and growth of organoids of constitutive NoxO1 and Nox4 knockout mice were not different from a wildtype control (**Figure 3D**), indicating than neither NoxO1 nor Nox4 are essential for proliferation in intestinal organoids. From Figure 1C, we further know that tamoxifen, in CreER^T2^ negative organoids is without effect on number and growth. We conclude, not deletion of NADPH oxidases or reduction of ROS, but induction of Cre-recombinase activity was the reason of impaired organoid growth in Figure 3C. This conclusion is even more supported by the fact that NoxO1 expression disappears within 48 h in the course of culturing wildtype intestine organoids [26] without affecting their viability or growth. Accordingly, instead of studying the role of a gene of interest in organoids, utilizing the gtRoasCreER^T2^ system may result in investigation of the effects of acute Cre-recombinase activation.

### 3.3. Acute Activation of Cre-Recombinase Causes DNA Damage and Induces Apoptosis

Impaired survival and proliferation of organoids can result from manifold reasons. As pointed out already, earlier work by others suggests off target effects of Cre recombinase due to similarities to the flox sequence within the genome of mouse and men. DNA damage was identified by alkaline comet assay as well as DAPI based analysis of cell cycle progression (**Figure 4A,B**).

Both assays verify that indeed, activation of Cre recombinase induces DNA double strand breaks and DNA fragmentation (Sub-G1 phase cells). DNA damage could reduce the expression of Lgr5 or Wnt3a. However, Lgr5 expression was not affected by 4-OHT (Appendix A).

Accordingly, we thought about a more general effect as underlying mechanism of impaired organoid growth. Especially in the light of an actual reduction of cell number as shown in Figure 1B, we analyzed for apoptosis in organoids. Using a PI/Annexin V assay, we found the percentage of healthy cells to be decreased upon treatment with 4-OHT and early stage apoptotic cells were significantly increased (**Figure 5A,B**). In contrast, gtRosaCre organoids did not show any DNA double strand breaks or apoptosis. We conclude that off target effects of acutely activated Cre-recombinase lead to DNA double strand breaks and DNA fragmentation, which in turn induce apoptosis of cells within organoids.

## 4. Discussion

Transgenic and knockout mice are essential to study the function of genes in mammalian tissue. The Cre/LoxP system is one of the most widely used gene-editing technologies to generate inducible mutations and knockouts in cells, tissue and whole animals [27]. Non-specific effects of Cre-recombinase are well described *in vivo* and *in vitro* [13]. Among those, recombination in unintended tissues or cells [27] and Cre-recombinase related toxicity due to damage of the DNA at pseudo-loxP sites [14] are the most problematic ones. Nevertheless, many studies are based on analysis of tissue or cells derived from Cre/loxP mice or retro virus based overexpression of Cre-recombinase in floxed tissue or cells [4]. Especially analysis of complex tissues as found in the gut heavily relies on *in vitro* methods such as generation of intestinal organoids [28].

In this study we present evidence that acute tamoxifen induced nuclear translocation of Cre-recombinase induces apoptosis and cell death due to massive DNA damage in isolated intestinal cryps, which eventually interferes with organoid formation. Although to our knowledge this effect has not been described for organoids so far, it is similar in fibroblasts [14] and keratinocytes [29]. In contrast, others using intestinal-specific Villin-CreERT2 animals and tamoxifen-induced activity of Cre-recombinase in organoids did not describe growth arrest or apoptosis [4,30,31]. As villin is mainly expressed in the villi compared to the crypts [32] it is possible that in such models the stem cells, which give rise to new organoids, do not express Cre-recombinase and therefore activation of the enzyme is not harmful. In fact, rapidly proliferating precursors are likely more susceptible to Cre-recombinase-mediated damage as activation of Cre-recombinase prevents proliferation and differentiation of multiple cells of hematopoietic lineage *in vivo* and *in vitro* [33]. In enteroids stem cells can be crypt-base columnar cells [34] as well as reserve stem cells at position 4 of the crypt [35]. In our study we isolated whole crypts as basis for organoid development and tamoxifen treatment started right after crypt isolation. Accordingly, further studies are needed to identify the cell or the cells mainly hit by Cre-recombinase driven apoptosis and growth inhibition. Nevertheless, the CreERT2 system was developed to achieve temporal regulation of transgenes without adverse effects of constitutive Cre-recombinase activity [33]. Our study indicates that at least in intestinal organoids, Cre-recombinase toxicity may give rise to false results as shown on the example of NADPH oxidases. This effect is likely to be less pronounced in organoids derived from mice overexpressing constitutively active Cre-recombinase, as live long exposure may trigger the development of an DNA repairing system [33].

Our study stresses that the inducible Cre/loxP system has unwanted effects that need to be addressed when designing a study. Unfloxed controls carrying the Cre-recombinase transgene or alternatives such as siRNAs, Crispr/CAS or knockouts are essential to discriminate off-target effects of Cre-recombinase from those of the gene of interest. Further, concentration and duration of 4-OHT induced Cre-recombinase activity needs to be optimized for each experiment. Although those actions require more controls and pretests, it is worthwhile doing so in order to limit the likelihood of analyzing effects attributed to Cre-recombinase toxicity or infidelity instead of the intended gene or protein of interest.

## 5. Conclusions

Time and tissue specific deletion of a gene of interest is an attractive approach if the knockout of the gene of interest is embryonically lethal or results in severe disabilities of the model animal. Often an inducible Cre-recombinase is used to dissect a gene or region of interest flanked by loxP sites. However, here we provide evidence that acute activation of Cre-recombinase can induce DNA damage, apoptosis and growth inhibition in intestinal organoids. Many of these effects could appear *in vivo* as well, and may wrongly being ascribed to the loss of the gene of interest. Accordingly, false results could arise from studies lagging proper controls. This is especially important if the aim of a study is to test drugs or to generate preclinical data for potential therapeutics. We recommend the use of controls as suggested above. To do so requires sufficient animal capacities, which may not be available to every investigator. Nevertheless, the possibility of unintended side effects of acutely induced Cre-recombinase activity should be taken into account, when interpreting results of a study and at least one additional control should be performed to show that the effects seen are consequence of deletion of the gene of interest.

## Figures and Tables

**Figure 1 antioxidants-11-01452-f001:**
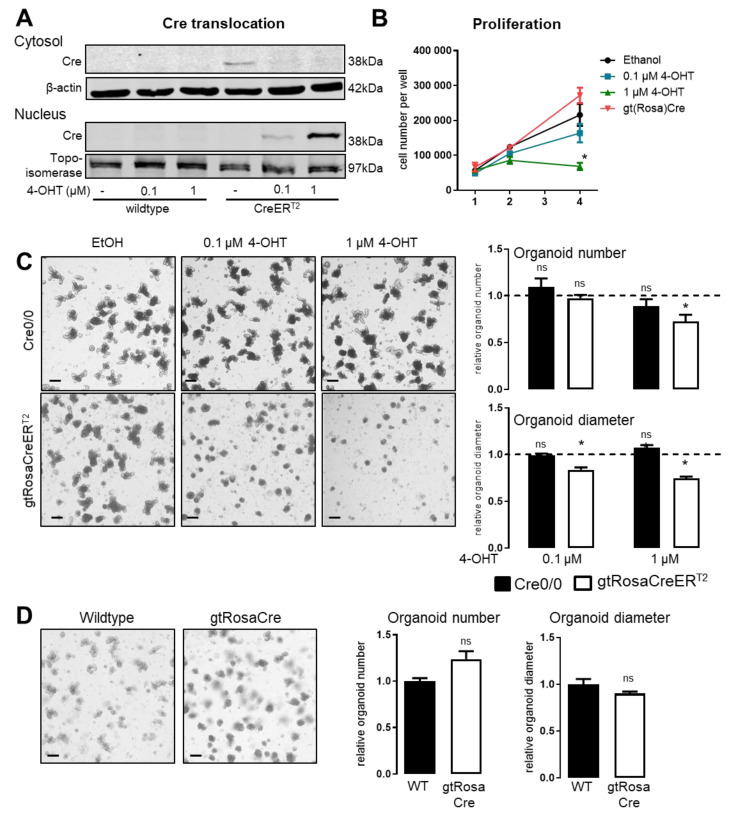
**4-OHT induced nuclear translocation of Cre-recombinase inhibits organoid growth.** (**A**) Nuclear extraction of proteins to analyze CreER^T2^ translocation from the cytosol to the nucleus after 48 h of 4-OHT treatment (**B**) Cell number per well of digested organoids at indicated time points. n = 7–8; mean with SEM * *p* < 0.05; 2-way Anova; (**C**,**D**) Organoids generated from mice with the following genotype: Cre0/0 and gtROSACreER^T2^ (**C**) or C57BL/6 (wildtype) and gtRosaCre (**D**) were analyzed for number and diameter on day 4 after isolation. n = 5–13; mean with SEM t-Test; ns: none significant and * *p* > 0.05 without vs. with 4-OHT in C, and WT vs. gtRosaCre in D, bars = 200 µm.

**Figure 2 antioxidants-11-01452-f002:**
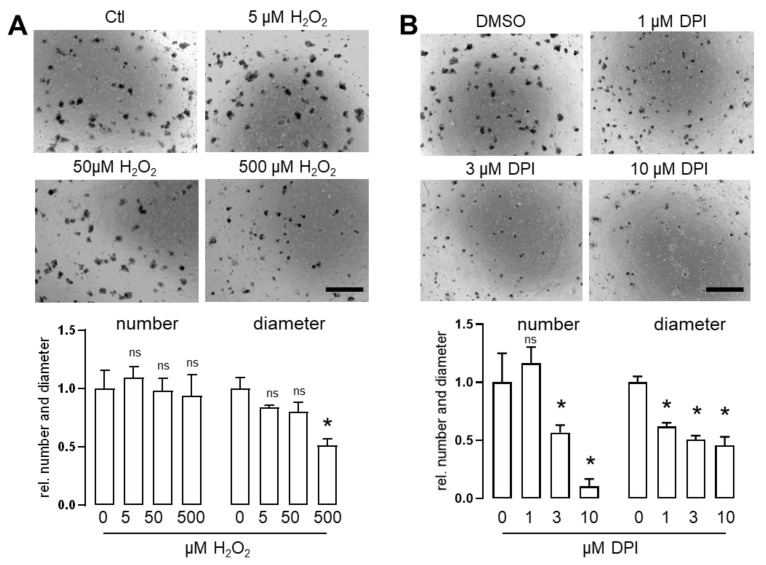
**Inhibition of flavoproteins leads to death of organoids.** (**A**,**B**) Small intestinal organoids from C57BL/6 mice were treated with either DMSO, 5 µM H_2_O_2_ (**A**) or 3 µM DPI (Diphenyleneiodonium) (**B**). Organoid number and diameter were measured as a marker for organoid survival and proliferation. Values were calculated relative to untreated control. n= 4–6; mean with SEM; ns: none significant * *p* < 0.05 ctl vs. treatment indicated (one-way Anova); bars = 500 µM.

**Figure 3 antioxidants-11-01452-f003:**
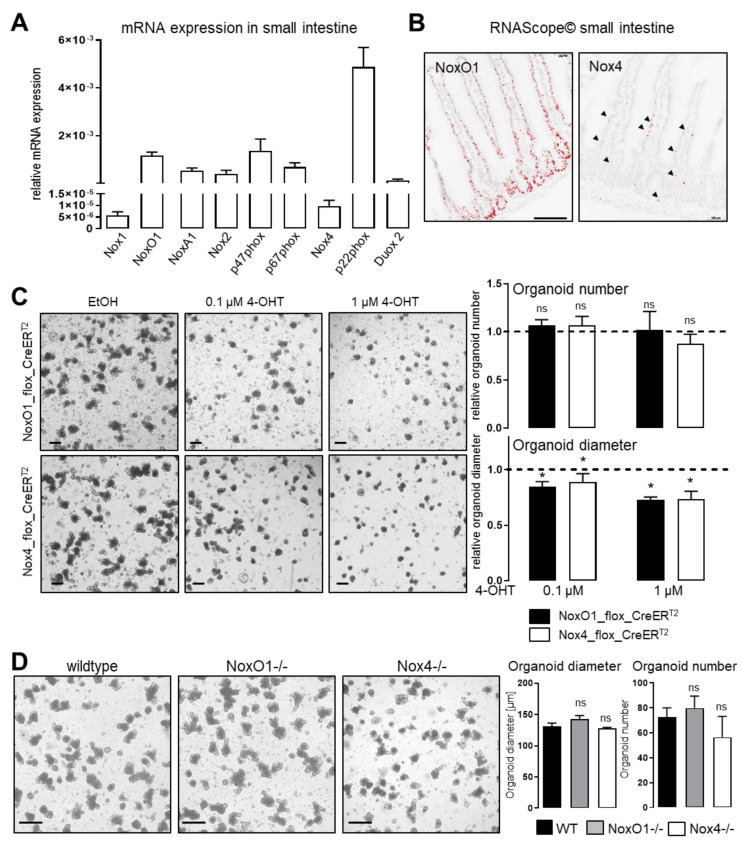
**Cre-recombinase, but not NADPH oxidases interfere with organoids growth.** (**A**) qRT-PCR for Nox subunits in small intestinal tissue. mRNA expression relative to EF2 housekeeping gene. n = 9; mean with SEM; (**B**) Detection of NoxO1 and Nox4 mRNA in small intestine by in situ hybridization. Scale bar 200 µm; (**C**) Treatment of small intestinal organoids with either Ethanol control or 0.1 or 1 µM of 4-OHT (4-Hydroxytamoxifen). Organoids derived from differently floxed CreERT2 mouse lines. Organoids were treated with 4-OHT for 48 h. Organoid number and diameter were analyzed on day 4 after isolation and normalized to EtOH treated control. Dashed lines indicate ethanol-treated controls n = 4–13; mean with SEM; ns: none significant and * *p* < 0.05 EtOH vs. 4-OHT (two-way Anova, Bonferroni post-test); (**D**) Small intestinal organoids from mice constitutively lacking NoxO1 or Nox4, compared to wildtype controls. ns: none significant WT vs. kockout as indicated (two-way Anova, Bonferroni post-test); n = 5–10; bars = 200 µm.

**Figure 4 antioxidants-11-01452-f004:**
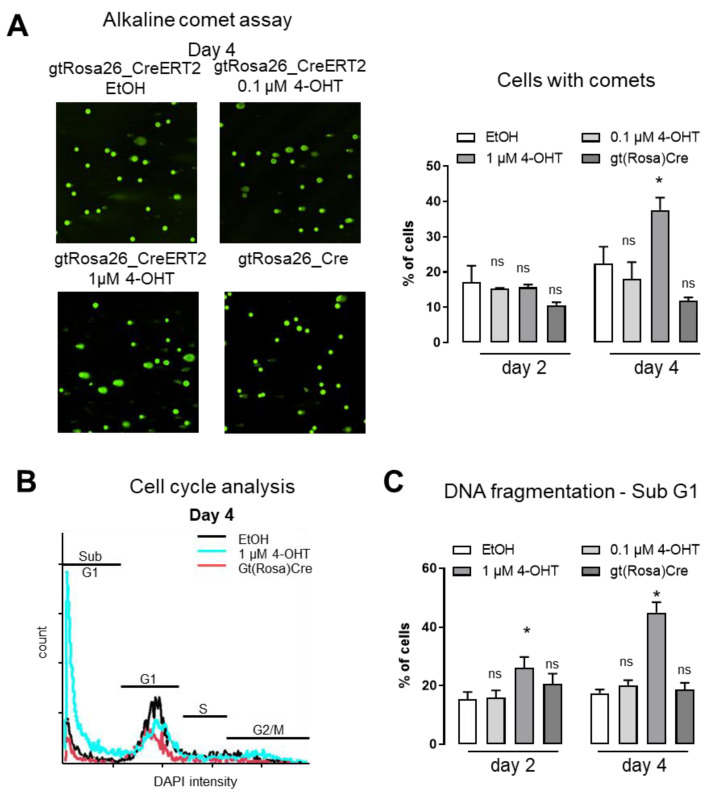
**Acute activation of Cre-recombinase forces DNA damage** (**A**) Alkaline Comet Assay stained with SYBR Gold. Left panel representative images. Percentage of cells with comets. * *p* < 0.05, n = 3–6; mean with SEM. 2-way Anova. (**B**) Representative raw data plot of DNA fragmentation and cell cycle analysis of DAPI stained cells using the Nucleocounter NC-3000 treated with 4-OHT as indicated. Cells were analyzed at day 4. Cells with fragmented DNA are in SubG1 phase. (**C**) Percentage of cells with fragmented DNA was analyzed. n = 6; mean with SEM; ns: none significant and * *p* < 0.05 EtOH vs. 4-OHT or gtRosaCre as indicated; 2-way Anova.

**Figure 5 antioxidants-11-01452-f005:**
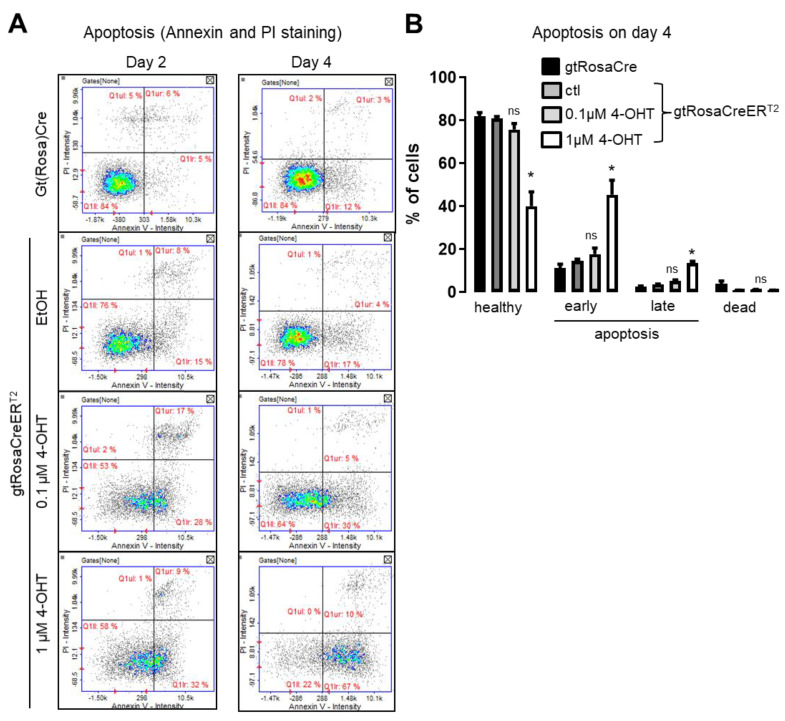
**4-OHT-induced Cre activity reduced cell numbers and induction of apoptosis.** (**A**) Annexin V Apoptosis Analysis using the Nucleo Counter NC-3000. Lower left panel depict healthy cells, lower right panel early apoptotic cells, upper right panel late apoptotic cells, upper left panel dead cells. (**B**) Annexin V Staining statistics of raw data. n = 6; mean with SEM; ns: none significant and * *p* < 0.05 EtOH vs. 4-OHT or gtRosaCre as indicated; 2-way Anova.

**Table 1 antioxidants-11-01452-t001:** Primer Sequences for qRT-PCR.

	Forward 3′-5′	Reverse 3′-5′
**m β-actin**	TGACAGGATGCAGAAGGAGA	GCTGGAAGGTGGACAGTGAG
**h,m,r EF2**	GACATCACCAAGGGTGTGCAG	GCGGTCAGCACACTGGCATA
**m Duox2**	TCTTCACCATGATGCGGTCC	GGAGTCCGGTTGATGAACGA
**m Nox1**	CCTCCTGACTGTGCCAAAGG	ATTTGAACAACAGCACTCACCAA
**m Nox4**	TTGTGTTAGGATCCGGGTTGT	GCTCCTAGTGTCTTCCAGGGA
**m NoxA1**	AGATACGGGACTGGCACCG	CATCCTAGCCAGCGGCTCTC
**m NoxO1**	ACTTAAACGCCTGTGCCATC	CCCCAACACTGCCCTAAGTA
**m p22phox**	TGTGGTGAAGCTTTTCGGGC	GGATGGCTGCCAGCAGATAGAT
**m p47phox**	TCCCAACTACGCAGGTGAAC	CCTGGGTTATCTCCTCCCCA
**m p67phox**	CTATCTGGGCAAGCCTACGGTT	CACAAAGCCAAACAATACGCG

## Data Availability

Data is contained within the article or Appendix A.

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
