# Peer review of "Cre-Recombinase Induces Apoptosis and Cell Death in Enterocyte Organoids"

_antioxidants, 2022, doi:10.3390/antiox11081452_

Round 1
Reviewer 1 Report
The authors have nicely summarized the study undertaken and presented it in a good way. Despite interesting work, authors should undertake minor revisions to make it appropriate for publication.
Figure 1. Authors need to indicate which two groups were compared.
Besides, if there is no significance, indicate it as "ns".
Figure 2. No p values. * P < 0.05 given in figure legend, but bars are missing it.
Figure 3: Here, A and D are missing p values.
Figure 4; again, same as above, better indicate which two groups are compared. If there is no significance, mark it as "ns".
Figure 5; again, same as above, better indicate which two groups are compared. If there is no significance, mark it as "ns".
The conclusion section needs to be a bit more elaborative and should highlight the importance of the study and future directions with possible limitations.
Author Response
Reviewer 1
Comments and Suggestions for Authors
The authors have nicely summarized the study undertaken and presented it in a good way. Despite interesting work, authors should undertake minor revisions to make it appropriate for publication.
Figure 1. Authors need to indicate which two groups were compared.
Besides, if there is no significance, indicate it as "ns".
Figure 2. No p values. * P < 0.05 given in figure legend, but bars are missing it.
Figure 3: Here, A and D are missing p values.
Figure 4; again, same as above, better indicate which two groups are compared. If there is no significance, mark it as "ns".
Figure 5; again, same as above, better indicate which two groups are compared. If there is no significance, mark it as "ns".
We thank the reviewer for careful reading of our manuscript and revised the figures as requested.
The conclusion section needs to be a bit more elaborative and should highlight the importance of the study and future directions with possible limitations
We extended the text of the conclusion section: “Time and tissue specific deletion of a gene of interest is an attractive approach if the knockout of the gene of interest is embryonically lethal or results in severe disabilities of the model animal. Often an inducible Cre-recombinase is used to dissect a gene or region of interest flanked by loxP sites. However, here we provide evidence that acute activation of Cre-recombinase can induce DNA damage, apoptosis and growth inhibition in intestinal organoids. Many of these effects could appear in vivo as well, and may wrongly being ascribed to the loss of the gene of interest. Accordingly, false results could arise from studies lagging proper controls. This is especially important if the aim of a study is to test drugs or to generate preclinical data for potential therapeutics. We recommend the use of controls as suggested above. To do so requires sufficient animal capacities, which may not be available to every investigator. Nevertheless, the possibility of unintended side effects of acutely induced Cre-recombinase activity should be taken into account, when interpreting results of a study and at least one additional control should be performed to show that the effects seen are consequence of deletion of the gene of interest.”
Reviewer 2 Report
The Paper “Cre-recombinase induces apoptosis and cell death in enterocyte organoids” reports important data about these systems.
Authors showed results related to NADPH oxidases in the organoid system. In addition data support that activation of Cre recombinase by 4-hydroxy tamoxifen in freshly isolated enteroids leading to decreased diameter of organoids. A dose-dependent effect was observed. Other data regard DNA damage and apoptosis.
I suggest minor revision:
Material and methods
Authors should provide more data for primary antibodies.
Results
Fig. 1 A WB analysis. In WB image the molecular weight of each analyzed protein should be reported. Also, was a quantitative analysis of the bands done? Is it possible report it?
C and D. A bar should be reported.
Figure 2 A and B. Bar is reported but its value is not present.
Similar problems are present in Fig 3 and 4
Authors should re-read the whole paper in order to eliminate repeated concepts and errors in sentence construction.
Authors should review the presentation of the figures in order to make them clearer.
Author Response
Reviewer 2
Comments and Suggestions for Authors
The Paper “Cre-recombinase induces apoptosis and cell death in enterocyte organoids” reports important data about these systems.
Authors showed results related to NADPH oxidases in the organoid system. In addition data support that activation of Cre recombinase by 4-hydroxy tamoxifen in freshly isolated enteroids leading to decreased diameter of organoids. A dose-dependent effect was observed. Other data regard DNA damage and apoptosis.
I suggest minor revision:
Material and methods
Authors should provide more data for primary antibodies.
We thank the reviewer for carefully reviewing our manuscript. The following information for primary antibodies is included in the reviced version of the manuscript: “Primary antibodies used are: D7L7L from Cell signaling for Cre-recombinase, A1978 from Sigma for β-actin and sc-5342 from Santa Cruz for Topoisomerase I.”
Results
Fig. 1 A WB analysis. In WB image the molecular weight of each analyzed protein should be reported. Also, was a quantitative analysis of the bands done? Is it possible report it?
Markers for molecular weight are now included into the figure. However, we have to apologize for no quantification, as figure 1 was just meant as a prove of principle for the well known system of 4-OHT mediated nuclear translocation of Cre-recombinase. We did this only twice and therefore no statistics can be performed. In oder to please the reviewer we include the original western blots below:
C and D. A bar should be reported.
Figure 2 A and B. Bar is reported but its value is not present.
Similar problems are present in Fig 3 and 4
We thank the reviewer for careful reading of our manuscript and revised the figures as requested.
Authors should re-read the whole paper in order to eliminate repeated concepts and errors in sentence construction.
The manuscript was corrected at several sentences. We hope the changes will satisfy the reviewers’ request.
Authors should review the presentation of the figures in order to make them clearer.
We followed the advice of reviewer 1, which potentially will help to make the figures clearer and hope that this will also satisfy this appeal of reviewer 2.
